# Does Poor Mental Health Impair the Effectiveness of Complementary Food Hygiene Behavior Change Intervention in Rural Malawi?

**DOI:** 10.3390/ijerph191710589

**Published:** 2022-08-25

**Authors:** Jurgita Slekiene, Kondwani Chidziwisano, Tracy Morse

**Affiliations:** 1Global Health Engineering (GHE), Department of Mechanical and Process Engineering (D-MAVT), ETH Zurich, Clausiusstrasse 37, 8092 Zurich, Switzerland; 2Centre for Water, Sanitation, Health and Appropriate Technology Development (WASHTED), Malawi University of Business and Applied Sciences (MUBAS), Private Bag 303, Chichiri, Blantyre 3, Malawi; 3Department of Environmental Health, Malawi University of Business and Applied Sciences (MUBAS), Private Bag 303, Chichiri, Blantyre 3, Malawi; 4Department of Civil and Environmental Engineering, University of Strathclyde, Level 5 James Weir Building, Glasgow G1 1XQ, UK

**Keywords:** behavior change, public health, humanitarian action, mental health, RANAS, rural Malawi, complementary food hygiene behaviors, evidence-based health promotion, intervention effectiveness

## Abstract

Mental disorders have the potential to affect an individual’s capacity to perform household daily activities such as water, sanitation, and hygiene (food hygiene inclusive) that require effort, time, and strong internal motivation. However, there is limited detailed assessment about the influence of mental health on food hygiene behaviors at household level. We conducted a follow-up study to detect the effects of mental health on food hygiene behaviors after food hygiene intervention delivery to child caregivers in rural Malawi. Face-to-face interviews, based on the Risk, Attitude, Norms, Ability, and Self-regulations (RANAS) model, were conducted with 819 participants (control and intervention group) to assess their handwashing and food hygiene-related behaviors. Mental health was assessed using the validated Self-Reporting Questionnaire. Study results showed a significant negative relationship between mental health and handwashing with soap behavior (*r* = −0.135) and keeping utensils in an elevated place (*r* = −0.093). Further, a significant difference was found between people with good versus poor mental health on handwashing with soap behavior (*p* = 0.050) among the intervention group. The results showed that the influence of the intervention on handwashing with soap behavior was mediated by mental health. Thus, integration of mental health in food hygiene interventions can result in improved outcomes for caregivers with poor mental health.

## 1. Introduction

Diarrheal disease is one of the leading causes of death among children under the age of five years in Low and Middle-Income Countries (LMICs) with 424,000 deaths annually [1]. Childhood diarrhea is high during the first two years of life which coincides with the introduction of complementary food [2]. Thus, among other factors, contaminated food has been linked to the onset of diarrhea among young children [3,4,5]. Such childhood diarrhea has been related to unhygienic food preparation and storage environments including the presence of animals in the household setting and lack of handwashing at critical times [6,7]. Previous research revealed the importance of household food hygiene behaviors on child health [4,8]. However, despite its relevance, uptake of food hygiene behaviors in the household domain remains low [9,10]. 

Access to safe water, improved sanitation facilities, hygienic weaning practices, and improved personal and household hygiene have been identified by the World Health Organization (WHO) and UNICEF as among the important factors supporting the reduction of childhood diarrhea [11]. Nevertheless, the presence of Water, Sanitation, and Hygiene (WASH) infrastructure alone does not result in improved WASH-related behaviors (including food hygiene) [12,13]. Thus, to improve hygiene behaviors, innovative, theory-based hygiene promotions need to be designed and assessed [14].

The Risk, Attitude, Norms, Ability, and Self-regulation (RANAS) model of behavior change was applied to detect factors associated with household food hygiene behaviors [15]. The RANAS model provides a detailed approach on how to identify behavioral factors influencing the performance of Water, Sanitation, and Hygiene (WASH) related behaviors. The use of the RANAS approach has proven to be successful in improving WASH and food hygiene behaviors at the household level in LMICs [16,17,18]. 

The RANAS model of behavior change includes five psychosocial factor blocks from various existing theories of psychology [15]. Risk factors present an understanding of a person’s level of knowledge on his/her susceptibility to diseases. Attitude factors refer to a person’s beliefs about the costs and benefits and feelings connected with the behavior. Normative factors relate to individual perception about the behavior of others, approval or disapproval of important others, and personal obligation of a particular behavior. Ability factors refer to a person’s capabilities to execute and maintain the desired behavior. Self-regulation factors highlight the person’s capacity to develop a plan on how to sustainably execute a behavior, which includes measures put in place to address the obstacles encountered during behavior performance. Further, the RANAS model considers contextual factors that may have an influence on an individual’s performance of the desired behavior. The RANAS model categorizes the contextual factors into: (1) Social factors which include the culture, economic and social relations; (2) Physical factors which include the natural and built environment; and (3) Personal factors which constitute demographic factors such as marital status and household size. 

Evidence suggests that mental disorders have the potential to affect an individual’s capacity to carry out daily activities [19]. Mental disorders can result from various social influences such as inadequate access to food, malnutrition, and diseases (e.g., HIV/AIDS) [20,21,22]. Thus, household WASH and food hygiene behaviors that require time, effort, and self-confidence may potentially be influenced by internal mental conditions such as stress, anxiety, and depression. For instance, depression remains a prevalent mental disorder in LMICs where it contributes to one’s inability to function effectively on a daily basis [23,24,25]. Importantly, maternal mental health has an influence on child health and development, potentially impairing cognitive, emotional, and behavioral outcomes [26,27,28]; and women have a higher susceptibility to mental disorders than men [29]. Importantly, WASH practices including food hygiene are mainly performed by women in LMICs [30]. 

Research in LMICs has highlighted the role of depression in negatively influencing specific WASH-related behaviors such as hand washing with soap, water collection, and latrine construction at the household level [31]. Importantly, it has been suggested that evidence-based interventions to improve mental health should be incorporated into WASH interventions [32]. However, based on our literature review, no detailed assessment about the influence of mental health on food hygiene-related behaviors at the household level have been conducted. 

Our study aims to detect the effects of mental health on complementary food hygiene behaviors after the intervention. The aim of our research is to provide practical implications to NGOs working in developing countries on why it’s necessary to integrate mental health in complementary food hygiene-related behavior change interventions. 

We hypothesized that poor mental health had a direct and indirect negative effect on hand washing with soap at key times (before eating main meals, before food preparation, before eating a snack, after using the toilet, and after cleaning a child’s bottom), keeping kitchen utensils in an elevated place, and washing kitchen utensils with soap and water after an intervention. We addressed the following research questions:

Is there a relationship between mental health and (a) handwashing with soap at key times, (b) keeping utensils in an elevated place, and (c) washing kitchen utensils with soap after intervention? 

Are there differences between people with good versus poor mental health in: (a) handwashing with soap at key times, (b) keeping utensils in an elevated place, and (c) washing kitchen utensils with soap after intervention? 

Does mental health mediate a relationship between intervention and (a) handwashing with soap at key times, (b) keeping utensils in an elevated place, and (c) washing kitchen utensils with soap after intervention? 

## 2. Materials and Methods

### 2.1. Study Area and Design

A longitudinal study design was applied to collect follow-up data from November to December 2018 from households that participated in an intervention and from households assigned to a control group. The study was conducted in rural Malawi; i.e., the Chikwawa district which has a population of 564,684 (of which 16% are younger than 5 years), [33] across 12 traditional authorities (TAs).

The study included 819 respondents (i.e., child caregivers) from three TAs. To ensure that there was uniformity in all the respondents included in the study, the following inclusion criteria were considered: Recruited respondent/household should be residing within the study areaPresence of a toilet in the householdThe household should access safe water from a borehole within a distance of 500 m.Presence of a child that was 4 and 90 weeks old to ensure that recruited children are not less than 4 weeks old and that the children will be aged five years and below during the end of project evaluation. To confirm the age and vaccination status of the recruited children, the enumerators verified the information from the child’s health passport book.

### 2.2. Data Collection Method

Data was collected by enumerators who were trained for five days prior to data collection. The training package of the enumerators included study objectives, data collection techniques, and translation of the questionnaire into a vernacular language (Chichewa). The training also highlighted the values of human research subjects during data collection. Validation of the data collection tool (i.e., the questionnaire) was done through pretesting. The pretesting process identified and addressed the challenges associated with administering the questionnaire which included revision of questions for easy understanding by the respondents. As a way of ensuring quality, the data collection process was supervised by one of the co-principal investigators.

### 2.3. Questionnaires and Measures

A questionnaire based on the RANAS model was used to conduct face-to-face interviews with the child caregivers. Specifically, the questionnaire collected self-reported data on demographic characteristics, mental health, how the participants were involved in the intervention, and psychosocial factors underlying handwashing and food hygiene behaviors. The questionnaire also included spot-check observations to assess the availability of WASH facilities. As indicated in Table 1, a 5-point rating scale (ranging from ‘not at all’ to ‘very much’) type of questions were used to capture information regarding the psychosocial factors associated with handwashing and food hygiene behaviors. 

Assessment of mental health was conducted with the use of the Self-Reporting Questionnaire (SRQ-20) that has been translated into Malawi’s vernacular language (Chichewa) [34]. A validation study reported that a score of ≥7 (score range 0–20) should be considered as a cutoff point [35]. Thus, the cutoff point between good and poor mental health groups was based on a score of 8 points and above were categorized as caregivers with poor mental health while those with 7 points and below as caregivers with good mental health [35]. Open Data Kit (ODK) platform (Department of Computer Science and Engineering, University of Washington, Seattle, WA, USA) on android tablets was used in capturing all data. 

### 2.4. Behavior Change Interventions

A formative research study targeting child caregivers guided the development and implementation of the ‘Hygienic Family’ behavior change (BC) intervention. The RANAS model of behavior change [15] guided the designing and delivery of the Hygienic Family intervention which aimed at improving food hygiene behaviors among child caregivers with children aged five years and below. Initially, a formative study was conducted to inform about the contextual and psychosocial factors that were to be considered during the design of the intervention. Cluster meetings and door-to-door household visits on alternating weeks were used as the main communicating channel for intervention delivery. The household visits reinforced the food hygiene behaviors that were initially introduced, demonstrated, and discussed during the cluster meetings. Mainly, the intervention was delivered by the female community volunteers with support from the government community health workers. More details on how the intervention was designed and delivered, and the results of the formative study including trial evaluation results have been published elsewhere [8,9,16,36,37]. The ‘Hygienic family’ BC intervention strategy was designed and developed by the SHARE research advisory group in Malawi with the support of the Swiss Federal Institute of Aquatic Science and Technology in Switzerland (EAWAG). 

### 2.5. Statistical Analysis of Data

The IBM SPSS 27 Statistics software and the PROCESS macro for SPSS were used to carry out the statistical tests (i.e., frequencies, correlations, *t*-test, and mediation analyses) [38]. The mean comparison analysis of contextual factors of study respondents (good versus poor mental health group) included the following contextual factors: gender, marital status, household size, income, wealth index (ownership of TV, mobile phone, bicycle, iron sheet, farming area, electricity, and running water), ownership of animals, and child diarrhea in past two weeks. Differences between people with good mental health and poor mental health in (a) handwashing with soap at key times, (b) keeping utensils in an elevated place, and (c) washing kitchen utensils with soap and water were calculated. A simple mediation model was computed using the PROCESS macro for SPSS 27 [38]. To estimate the confidence intervals of indirect effects, bootstrapping with 10,000× resampling was used. Three simple mediation models were tested where the intervention was included as a predictor, mental health as a mediator, and (a) hand-washing behavior before food preparation; (b) keeping utensils in an elevated place and washing kitchen utensils with soap and water as outcomes. 

The indirect (a*b), direct (c′), and total effects (c) of mental health on target behaviors were calculated. Path a estimates the effect of the intervention (predictor) on the mediator (mental health), and path b estimates the effect of the mediator (mental health) on target behavior (outcome). An indirect effect is the effect of intervention via mental health on target behavior. The direct effect is the effect of the intervention on target behavior independent of mental health. The total effect is the sum of the indirect effect and the direct effect. 

### 2.6. Ethics 

Ethics approval for the study was obtained from the University of Malawi’s College of Medicine Research Ethics Committee (P.04/16/1935). In addition, the local authorities in the Chikwawa district provided permission to conduct the research study. Participants’ written informed consent was obtained before inclusion in the study. Participants were provided with a unique identifying number, and data were anonymized during data analysis. The study was registered with the Pan African Clinical Trials Registry (PACTR201703002084166).

## 3. Results

### Sample Characteristics and Prevalence of Mental Disorders

Out of the 819 study respondents, 634 were assigned to the intervention group and 185 to the control group. Out of the 634 caregivers from the intervention group *N* = 100 (15.8%) reported poor mental health. Out of the 185 caregivers from the control group *N* = 36 (19.5%) reported poor mental health.

The *t*-test means comparison analysis of contextual factors showed no significant differences in household size and wealth index. The Chi-square test indicated significant differences between people with good compared with poor mental health in the intervention group (*p* = 0.000) in the presence of child diarrhea in the past two weeks, but no difference in the control group. There were no differences in marital status, income, and animal ownership in both groups, intervention, and control (Table 2).

In research question one; is there a relationship between mental health and (a) handwashing with soap at key times (b) keeping utensils in an elevated place, and (c) washing utensils with soap after intervention? we applied Pearson correlation analysis (Table 3).

For the intervention group the results revealed a significant negative relationship between mental health (continues variable, 1–20) and (a) handwashing with soap at key times *(r* = −0.135 **) and (b) keeping utensils in an elevated place (*r* = −0.093 *). For the control group, our results revealed significant negative correlations between mental health and (c) washing utensils with soap *(r* = −0.177 *). The levels of handwashing with soap at key times and keeping utensils on elevated place behaviors’ performance in the intervention group was lower among people with poor mental health compared to people with good mental health. The levels of washing kitchen utensils with soap in the control group were lower among people with poor mental health compared to their counterparts with good mental health.

To answer our second research question, are there differences between people with good versus poor mental health on (a) handwashing with soap at key times, (b) keeping utensils in an elevated place, and (c) washing kitchen utensils with soap after the intervention, we applied frequencies, *t*-test, and Cohen’s effect size analyses (see Table 4).

Statistical analysis revealed significant differences in the intervention group between people with poor mental health and those with good mental health on handwashing with soap at key times (*p* = 0.050), but no significant differences in (b) keeping utensils in an elevated place, and (c) washing kitchen utensils with soap. This means that people with good mental health reported higher handwashing with soap at key times performance after the intervention. No significant differences were detected in the control group. However, Cohen’s related population effect size calculations showed small effects in handwashing with soap and keeping kitchen utensils in an elevated place (*d* = 0.21) among the intervention group, which refers to the magnitude of a result [39]. 

To answer our third research question, does mental health mediate the relationship between the intervention and (a) handwashing with soap at key times, (b) keeping utensils in an elevated place, and (c) washing kitchen utensils with soap? we used a mediation analysis method with PROCESS for SPSS 27. To test the first mediation model with (a) handwashing at key times (continuous variable) mental health (continuous variable) was included as a mediator. Intervention (yes/no) was included as a predictor, and handwashing before food preparation was included as an outcome in a simple mediation model. Estimation of indirect effect simple mediator model with mental health as mediator calculated underlying mechanisms, the effect of exposure on the outcome that works through the mediator (Figure 1).

Indirect (a*b), direct (c′), and total effects (c) of intervention on handwashing behavior at key times were tested. The indirect effect (a*b) is the effect of intervention compared to no intervention condition (predictor) via mental health (mediator) on handwashing with soap at key times (outcome). The mediation results revealed a significant indirect effect on mental health (b = 0.0205, CI 0.0035 to 0.0439). These results explained the underlying relationship between intervention and handwashing with soap at key times. The influence of the intervention on handwashing with soap at key times was mediated by mental health. The direct effect (c′) quantifies the effect of the intervention on handwashing with soap independent of the effect of the mediator (i.e., mental health) on the outcome. This direct effect was significant (*p* = 0.000). The total effect (c), the sum of the direct effect, and the indirect effect of the intervention on handwashing with soap were also significant (*p* = 0.000). 

However, contrary to our expectations, mediation models with target behaviors (b) keeping utensils in an elevated place, and (c) washing kitchen utensils with soap were not significant.

## 4. Discussion

This research study aimed at detecting the influence of mental health on complementary food hygiene behaviors after the intervention. The aim of our research was to provide practical implications to NGOs working in developing countries on why it’s necessary to integrate mental health in complementary food hygiene-related interventions. 

According to study findings, 16.6% of caregivers were at risk of developing common mental health disorders (CMD) which is in line with the previous research from sub-Saharan Africa [34,40]. Our study results suggest a significant association between caregivers’ mental health and child diarrhea in the past two weeks in the intervention group, which confirms previous findings from Malawi examining maternal CMD and infant growth [41]. 

Our first research question was whether there is a relationship between mental health and (a) handwashing with soap at key times, (b) keeping kitchen utensils in an elevated place, and (d) washing kitchen utensils with soap after behavior change intervention. In line with our expectations, we found significant negative associations in the intervention group between poor mental health and two complementary food hygiene behaviors (handwashing with soap at key times and keeping utensils in an elevated place). Caregivers with poor mental health performed handwashing at key times and keeping utensils on elevated place behaviors after the intervention less than people with good mental health. The study demonstrated an association between caregivers’ CMD and two food hygiene-related behaviors. This confirms findings from previous research studies on handwashing with soap in Zimbabwe and Malawi [32,42]. The present findings are in line with findings from rural China reporting associations between depressive symptoms and poor hand hygiene [43]. Previous research from 11 countries in Asia and Africa suggests that individuals with poor mental health status are more likely to report poor hygiene behaviors than individuals with good mental status [44]. Our study findings confirmed the negative association) between mental health and two food hygiene-related behaviors after an intervention.

Our second research question was on whether there were differences between people with good versus poor mental health in target behaviors: (a) handwashing with soap at key times; (b) keeping utensils in an elevated place, and (c) washing kitchen utensils with soap after interventions and comparing with a control group. In line with our expectations, study results revealed significant differences between people with good mental health and those with poor mental health on handwashing with soap before food preparation. Among people with good mental health, handwashing with soap before food preparation increased more after the intervention compared to those with poor mental health. This is in line with previous research on changes in handwashing behavior in rural Malawi [42]. Contrary to our expectations we could not find significant differences in two other target behaviors, namely, keeping kitchen utensils in an elevated place and washing kitchen utensils with soap after an intervention. Possible reasons will be discussed in the section below. No differences in all target behaviors between the poor and good mental groups were demonstrated in the control group, which could be explained by very low behavioral frequencies in both groups (poor vs good mental health) compared to intervention groups (poor vs good mental health).

Our third research question was whether mental health mediated the relationship between the intervention (yes = intervention group), no = control group) and (a) handwashing with soap at key times, (b) keeping utensils in an elevated place, and (c) washing kitchen utensils with soap after an intervention. Only the mediation model which included handwashing with soap at key times was significant. Mental health mediated the relationship between the intervention and handwashing with soap at key times. Our study results suggest that having poor mental health negatively predicted handwashing with soap at key times after implementation of the intervention. Nevertheless, the results also suggest that even after accounting for the mediating role of poor mental health, the implemented behavior change intervention still had a positive impact on handwashing at key times—post-intervention. Additionally, mediation results showed that the hygiene behavior change intervention had already had a positive impact on mental health which could be explained by the positive ‘side effects’ of participation in an intervention [45]. Contrary to our assumptions, mediation analysis was not significant for the other two food hygiene-related target behaviors: keeping kitchen utensils in elevated places and washing kitchen utensils with soap. 

Psychological research suggests that behavior includes not only physical activity, but also represents cognitive and affective processes that in turn guide the behavioral outcome. Behavior performed via cognitive processes was suggested by the Theory of Planned behavior [46]. The theory proposed that people logically evaluate available information. Current research suggests the importance of affective processes in health-related behavior change [47,48,49], e.g., well-being that consists of pleasure and happiness [49]. Furthermore, healthy emotion regulation plays a key role in good mental health and is impaired in people with poor mental health [42,50]. The present study findings suggest that handwashing with soap at key times is closely related to the caregiver’s mental condition. This is in line with recent research findings, that handwashing emotions mediate the relationship between messaging and handwashing intention, meaning that feelings about handwashing (in this case related to COVID-19 prevention) increased handwashing intention [51]. The possible reason why in our study poor mental health negatively influenced only handwashing with soap may be an individual’s affective/emotional processes by executing handwashing behavior, which is in line with our baseline study findings confirming that affective beliefs (feelings) predicted handwashing at key times behavioral performance [36]. Thus, handwashing with soap behavior was mediated by mental health compared to the other two food hygiene-related behaviors. 

Our study results confirmed that the impact of the intervention was affected by poor mental health which in turn led to lower handwashing with soap at key times. Poor mental health hinders the influence of the intervention in the behavior change process and highlights the need to consider mental health in complementary food hygiene interventions and include child caregivers with poor mental health in humanitarian action. However, it should also be noted that hygiene intervention alone, when compared to the control group, already showed positive effects on the mental health of vulnerable people which may be related to the method of implementation and development of social capital [45].

### Practical Implications

This study highlights the important role of mental health in the hygiene behavior-change process. An integrated approach targeting mental health and food hygiene interventions, a key but often neglected area of the F diagram [52], would be more effective. To improve household food hygiene effectively, mental health treatment should be incorporated into food hygiene interventions. Specifically, as reported elsewhere [42], our study confirms that handwashing with soap intervention should contain mental health messaging to strengthen the resilience of vulnerable caregivers.

The existing food hygiene intervention strategies should be refined according to the identified significant mediator influencing the targeted food hygiene behavior. Our study results suggest that mental health was a significant mediator for the handwashing with soap behavior before food preparation. Thus, specific mental health targeting interventions with the guidance of the RANAS catalog of the BCTs would be effective in improving handwashing with soap before food preparation, especially among child caregivers with poor mental health. This would increase complementary food hygiene behavioral outcomes that would result in childhood diarrheal disease reduction. The interventions could be delivered through group meetings as this has been proven to be successful in the implementation of mental health therapies among vulnerable populations in LMICs [53,54,55]. Importantly, the findings of the study emphasize the need for the inclusion of community members with poor mental health in WASH interventions.

The study has shown that mental health had no significant influence on washing utensils with soap and keeping utensils in an elevated place behavior. Thus, mental health activities would not be included in an intervention to promote these two behaviors. This requires further review in future research. Furthermore, the study was conducted in a rural area of southern Malawi, further research should compare rural and urban areas.

## 5. Conclusions

Integration of mental health interventions in hygiene interventions can result in improved outcomes for caregivers with poor mental health. This is especially relevant for behavioral outcomes, such as handwashing with soap, because mental health and emotions play important role in this behavior change process. Inclusion of child caregivers in humanitarian action will lead to better mental health among the vulnerable population, which in turn will increase hygiene-related behavior performance and health-related outcomes of child caregivers and their children.

## Figures and Tables

**Figure 1 ijerph-19-10589-f001:**
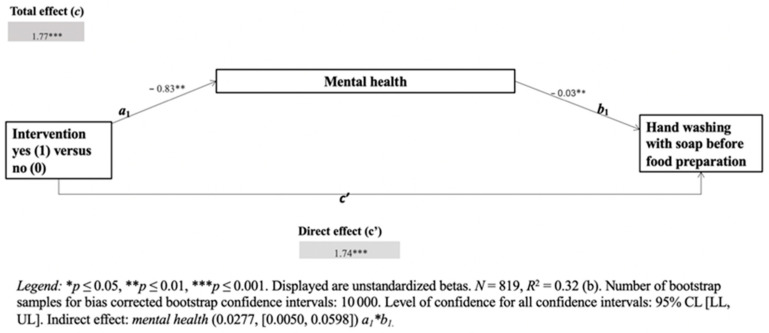
Mediation analysis: effects of intervention on handwashing at key times via mental health (mediator).

**Table 1 ijerph-19-10589-t001:** Key behavior questions.

Behaviors	Items	Answer Format
Hand washing before eating main meals (e.g., lunch) Hand washing after using the toilet Hand washing before food preparation Hand washing before eating snacks Hand washing after cleaning a child’s bottom	Before you eat main meals (e.g., lunch), how often do you wash your hands with soap and water? After you defecate, how often do you wash your hands with soap and water? Before you prepare food, how often do you wash your hands with soap and water? Before you eat snacks, how often do you wash your hands with soap and water? After cleaning a child’s bottom, how often do you wash your hands with soap and water	(Almost) at no time—(Almost) each time (5-point rating scale)
Washing kitchen utensils with soap	Before you use kitchen utensils, how often do you wash them with soap and water	(Almost) at no time—(Almost) each time (5-point rating scale)
Keeping kitchen utensils in an elevated place)	Do you keep your kitchen utensils in an elevated place	Not at all—very much (5-point rating scale)

**Table 2 ijerph-19-10589-t002:** Mean comparison of contextual factors of the study participants on mental health condition: poor versus good; *t*-test, Chi-square.

Behavior	Good MH *N* = 534 M (*SD*); %	Poor MH *N* = 100 M (*SD*); %	*p* Value	Good MH *N* = 149 M (*SD*); %	Poor MH *N* = 36 M (*SD*); %	*p* Value
	Intervention Group			Control Group		
Marital status Scale: Yes/No (married = 1, others = 0)	89% (yes)	87% (yes)	ns	89% (yes)	94% (yes)	ns
Household size	5.39 (1.94)	5.73 (1.84)	ns	5.08 (1.73)	5.50 (1.65)	ns
Income (MWK: Malawi Kwacha) 0–9999 (income low = yes)	38% (yes)	47% (yes)	ns	27.5% (yes)	14% (yes)	ns
Wealth Index (TV, mobile phone, farming area, iron sheets, bicycle, electricity, running water) Scale: Yes/No; sum scale: min. 0 to max. 7	1.92 (1.25)	2.03 (1.20)	ns	2.82 (1.35)	2.56 (1.46)	ns
Animal ownership Scale: Yes/No	65% (yes)	62% (yes)	ns	51% (yes)	47% (yes)	ns
Child diarrhea past two weeks *** Scale: Yes/No	4% (yes)	15% (yes)	0.000	16% (yes)	16.5% (yes)	ns

Note. * *p* ≤ 0.05, ** *p* ≤ 0.01, *** *p* ≤ 0.001. Intervention group *N* = 634; control group *N* = 185. ns = not significant.

**Table 3 ijerph-19-10589-t003:** Pearson correlation of target behaviors with mental health condition of the study participants: intervention and control group.

Behavior	Intervention Group *N* = 634, *r*	Control Group *N* = 185, *r*
Handwashing with soap at key times (factor)	−0.135 **	ns
Keeping utensils in an elevated place	−0.093 *	ns
Washing kitchen utensils with soap	ns	−0.177 *

Note. * *p* ≤ 0.05, ** *p* ≤ 0.01, *** *p* ≤ 0.001. Intervention group *N* = 634, control group *N* = 185. Cohen’s *d*, small: *d* = 0.20, medium: *d* = 0.50, large: *d* = 0.80. Scale for target behaviors: from 1 (not at all) to 5 (always); handwashing with soap at key times: combined variable (handwashing with soap before eating main meal, handwashing with soap before food preparation, handwashing with soap before eating a snack, handwashing with soap after using the toilet, handwashing with soap after cleaning child’s bottom); Scale for mental health yes/no (1–20). ns = not significant.

**Table 4 ijerph-19-10589-t004:** Mean comparison of target behaviors on the mental health condition of the study participants: poor versus good in intervention and control group.

Behavior	Good MH *N* = 534 M (*SD*)	Poor MH *N* = 100 M (*SD*)	*p*	Effect Size *d*	Good MH *N* = 149 M (*SD*)	Poor MH *N* = 36 M (*SD*)	*p*	Effect Size *d*
	Intervention Group				Control Group			
Handwashing at key times *	4.41 (0.64)	4.27 (0.79)	0.050	0.21	2.91 (1.08)	2.85 (0.84)	ns	0.06
Keeping utensils in an elevated place	4.65 (0.82)	4.46 (1.18)	ns	0.21	2.03 (1.60)	2.22 (1.71)	ns	−0.12
Washing kitchen utensils with soap	4.54 (0.73)	4.50 (0.75)	ns	0.06	3.83 (1.13)	3.61 (1.05)	ns	0.19

Note. * *p* ≤ 0.05, ** *p* ≤ 0.01, *** *p* ≤ 0.001. Intervention group *N* = 634, control group *N* = 185. Cohen’s *d*, small: *d* = 0.20, medium: *d* = 0.50, large: *d* = 0.80. Scale for target behaviors: from 1 (not at all) to 5 (always). Mental health: binary variable, good = 0, poor = 1; cutoff point ≥ 7. ns = not significant.

## Data Availability

The data presented in this study are openly available in PURE at [https://doi.org/10.15129/23d6e2d9-ef61-433e-9193-6a313085b531] (accessed on 29 April 2022).

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
