# Peer review of "Does Poor Mental Health Impair the Effectiveness of Complementary Food Hygiene Behavior Change Intervention in Rural Malawi?"

_ijerph, 2022, doi:10.3390/ijerph191710589_

Round 1

Reviewer 1 Report

This article is difficult to follow. First of all points: a, b, c, d must be clarify throughout the whole manuscript:

a) handwashing with soap at key times, b) keeping utensils on elevated place, and d) washing kitchen utensils with soap after intervention – line 103-117, 304,

or a) handwashing with soap at key times, b) keeping utensils on elevated place, and c) washing utensils with soap after intervention – line 222-226, 240-243, 288,

What is the difference between the intervention group and the control group?

Table 2 shows people with good and poor mental health in the control group (N = 149, N = 36, respectively). Why is there nothing about the control group in the "Sample characteristics and prevalence of mental disorders"?

Line 324: ... "after the implementation of the intervention" - what intervention do you think about?

Author Response

This article is difficult to follow. First of all points: a, b, c, d must be clarify throughout the whole manuscript:

a) handwashing with soap at key times, b) keeping utensils on elevated place, and d) washing kitchen utensils with soap after intervention – line 103-117, 304,

or a) handwashing with soap at key times, b) keeping utensils on elevated place, and c) washing utensils with soap after intervention – line 222-226, 240-243, 288, 

We made a correction, it’s a), b) and c)

What is the difference between the intervention group and the control group?

We added requested information.

The t-test mean comparison analysis of contextual factors showed no significant differences in household size and wealth index. Chi-square test indicated significant differences between people with good compared with poor mental health in intervention group (p=.000) in the presence of child diarrhea in the past two weeks, but no difference in control group. There were no differences in marital status, income, and animal ownership in both groups, intervention and control (Table 2). (line 217-219)

Table 2 shows people with good and poor mental health in the control group (N = 149, N = 36, respectively). Why is there nothing about the control group in the "Sample characteristics and prevalence of mental disorders"?

We added requested information (line 209).

Out of the 185 caregivers from control group N=36 (19.5%) reported poor mental health.

Line 324: ... "after the implementation of the intervention" - what intervention do you think about?

Paragraph 2.4 Behavior change intervention (line 159-170)

Reviewer 2 Report

This manuscript discussed that if poor mental health impaired the effectiveness of complementary food hygiene behavior change intervention in rural Malawi? This study highlights the important role of mental health in hygiene behavior change process, and it has academic value.

Please refer to the following comments and make corrections.

Page 3:

The selected questionnaire interview area Chikwawa is in the rural south of Malawi. Could the results be different if the study was done in the city?

Page 4:

7 points are used as the boundary, and descriptions above 7 points and below 7 points include 7 points, and it is recommended that they should be clearly distinguished.

The mediator is mental health, and is the confounder fully listed?

Page 9:

In clinical psychology, one of the symptoms of obsessive-compulsive disorder patients is excessive cleaning of their hands or the utensils they use. Is it possible that the scores of such patients may be biased?

Author Response

Page 3:

The selected questionnaire interview area Chikwawa is in the rural south of Malawi. Could the results be different if the study was done in the city?

Yes, we included this as a limitation (line 392-393)

Page 4:

7 points are used as the boundary, and descriptions above 7 points and below 7 points include 7 points, and it is recommended that they should be clearly distinguished.

The mediator is mental health, and is the confounder fully listed?

We made requested correction (line 153-155).

A validation study reported that a score of ≥7 (score range 0-20) should be considered as a cutoff point Thus, the cutoff point between good and poor mental health groups was based on a score of 8 points and above were categorized as caregivers with poor mental health while those with 7 points and below as caregivers with good mental health.

Page 9:

In clinical psychology, one of the symptoms of obsessive-compulsive disorder patients is excessive cleaning of their hands or the utensils they use. Is it possible that the scores of such patients may be biased?

To our knowledge there is no data from Malawi about the prevalence of obsessive-compulsive handwashing. However, the global lifetime OCD prevalence is 2-3% (Banerjee, 2020). However, execive handwashing is one possible symptom among others by OCD patients. Our study was conducted before COVID-19 pandemics with a sample size of total 819 participants. 3% from 819 would account for 24 participants. However,  we are looking at differences before and after intervention in handwashing scores (T1, intervention, T2).

Banerjee DD. The other side of COVID-19: Impact on obsessive compulsive disorder (OCD) and hoarding. Psychiatry Res. 2020 Jun;288:112966. doi: 10.1016/j.psychres.2020.112966. Epub 2020 Apr 11. PMID: 32334276; PMCID: PMC7151248.

Reviewer 3 Report

The submitted research aimed at detecting the influence of mental health on complementary food hygiene behaviours after the intervention. The findings of the research will be very helpful in designing strategies of health interventions in developing countries. I suggest following improvements in the manuscript.

Title of the manuscript is very lengthy. Make it concise and focused, good to have 12-18 words long.

Line 21: RANAS – Write full form at first instance.

Line 26: r=-.135, write r=-0.135. Follow same pattern everywhere.

Line 45: Repetition of reference 4.

References from 38 to 41 are not cited in text.

What was the basis for classifying mental health as good or poor?

Write in detail about which parameters considered under direct and indirect effect.

Describe which interventions were made on intervention group.

Author Response

Title of the manuscript is very lengthy. Make it concise and focused, good to have 12-18 words long.

Title of the manuscript has been revised to 17 words (line 2 – 3).

Line 21: RANAS – Write full form at first instance.

RANAS has been written in full at first instance (line 20 – 21).

Line 26: r=-.135, write r=-0.135. Follow same pattern everywhere.

Line 45: Repetition of reference 4.

Comment addressed accordingly (Line 25, Line 235, Line 237) and all tables

Repetition of reference 4 addressed (line 44).

References from 38 to 41 are not cited in text.

We have verified that these references are cited in text in the following lines:

Reference 38: Line 180 and 187

Reference 39: Line 259

Reference 40: Line 294

Reference 41: Line 296

What was the basis for classifying mental health as good or poor?

A validation study reported that a score of ≥7 (score range 0-20) should be considered as a cutoff point Thus, the cutoff point between good and poor mental health groups was based on a score of 8 points and above were categorized as caregivers with poor mental health while those with 7 points and below as caregivers with good mental health.

Write in detail about which parameters considered under direct and indirect effect.

The indirect effect (a*b) is the effect of intervention comparing to no intervention condition (predictor) via mental health (mediator) on handwashing with soap at key times (outcome). (line 272-274)

The direct effect (c′) quantifies the effect of the intervention on handwashing with soap independent of the effect of the mediator (i.e. mental health) on outcome. (line 278-279)

Describe which interventions were made on intervention group.

A brief description of the intervention has been included (Line 161 to 172). Further, references with more details about the intervention has been provided (Line 174).